# Interaction of Graphene Oxide Modified with Linear and Branched PEG with Monocytes Isolated from Human Blood

**DOI:** 10.3390/nano12010126

**Published:** 2021-12-30

**Authors:** Pavel Khramtsov, Maria Bochkova, Valeria Timganova, Anton Nechaev, Sofya Uzhviyuk, Kseniya Shardina, Irina Maslennikova, Mikhail Rayev, Svetlana Zamorina

**Affiliations:** 1Institute of Ecology and Genetics of Microorganisms UB RAS, 13 Golev str., 614081 Perm, Russia; krasnykh-m@mail.ru (M.B.); timganovavp@gmail.com (V.T.); kochurova.sofja@yandex.ru (S.U.); shardinak@gmail.com (K.S.); i.maslennikova1974@gmail.com (I.M.); mraev@iegm.ru (M.R.); mantissa7@mail.ru (S.Z.); 2Institute of Technical Chemistry UB RAS, 3 Academician Korolev str., 614013 Perm, Russia; nechaev.a@itcras.ru

**Keywords:** graphene oxide, immune cells, monocytes, viability, polyethylene glycol

## Abstract

Multiple graphene-based therapeutics have recently been developed, however potential risks related to the interaction between nanomaterials and immune cells are still poorly understood. Therefore, studying the impact of graphene oxide on various populations of immune cells is of importance. In this work, we aimed to investigate the effects of PEGylated graphene oxide on monocytes isolated from human peripheral blood. Graphene oxide nanoparticles with lateral sizes of 100–200 nm and 1–5 μm were modified with linear and branched PEG (GO-PEG). Size, elemental composition, and structure of the resulting nanoparticles were characterized. We confirmed that PEG was successfully attached to the graphene oxide surface. The influence of GO-PEG on the production of reactive oxygen species (ROS), cytokines, phagocytosis, and viability of monocytes was studied. Uptake of GO-PEG by monocytes depends on PEG structure (linear or branched). Branched PEG decreased the number of GO-PEG nanoparticles per monocyte. The viability of monocytes was not altered by co-cultivation with GO-PEG. GO-PEG decreased the phagocytosis of Escherichia coli in a concentration-dependent manner. ROS formation by monocytes was determined by measuring luminol-, lucigenin-, and dichlorodihydrofluorescein-dependent luminescence. GO-PEG decreased luminescent signal probably due to inactivation of ROS, such as hydroxyl and superoxide radicals. Some types of GO-PEG stimulated secretion of IL-10 by monocytes, but this effect did not correlate with their size or PEG structure.

## 1. Introduction

Graphene is a 2D carbon allotrope consisting of sp^2^ hybridized carbon atoms arranged in a honeycomb-like hexagonal lattice. Possessing a plethora of properties that are highly desirable for biomedical applications, graphene and its derivatives have attracted an enormous amount of attention from researchers [1]. Graphene oxide (GO) is an oxidized form of graphene containing carboxyl, hydroxyl, carbonyl, and epoxy groups on the edges of the lattice [2]. Polar functional groups endow graphene oxide with colloidal stability in water and provide sites for covalent functionalization.

Given the development of nanomedicine, it has become especially urgent to study the effects of nanomaterials in general and nanoparticles in particular on the function of cells of the immune system [3]. Graphene oxide can act as a scavenger of reactive oxygen species (ROS) [4] and has been applied to manipulate ROS-dependent polarization of macrophage cells in the site of inflammation [5]. Due to the ability of GO to change immune response polarization, cytokine production, and secretion of antibodies, it is broadly considered as a component of adjuvants [6,7] and immunotherapeutics [8].

Application of GO in vivo often requires its parenteral administration, which inevitably leads to interaction with immune cells in the bloodstream or the tissue fluid. Many recent studies shed some light on alteration of an immune response upon GO administration, however, effects of GO on some classes of immune cells, such as monocytes, dendritic cells, NK-cells, and so on, are still poorly understood [7,9]. In this regard, our study aims at the assessment of the viability and function of monocytes upon interaction with GO.

Monocytes, together with neutrophils, are important cells of innate immunity that protect the body from pathogenic microbes. The main functions of monocytes are phagocytosis, antigen presentation, and immunomodulation [10,11]. Being professional phagocytes, monocytes play a significant role in the removal of dead cells and cellular debris as well as in pathogen elimination [12,13]. Monocytes and monocyte-derived macrophages are actively involved in pathological processes in diseases such as cancer, atherosclerosis, arthritis, and inflammatory lung diseases [14]. Research is currently underway to target nanoparticles to these cells to diagnose and treat the above diseases [14].

Size [15] and coating [16] of GO both impact its uptake and alteration of cell physiology. Pristine GO has little potential to be utilized in clinics, because of its poor colloidal stability in saline and high cytotoxicity [17]. Modification with various polymers can significantly improve both stability and biocompatibility of GO. Polyethylene glycol (PEG) is a biocompatible polymer and a component of multiple nanoformulations approved for both clinical application and for undergoing clinical trials [18]. Indeed, recent studies have reported improved biocompatibility of PEG-modified GO in comparison with unmodified counterparts [17]. Given the above information, in this work, we examined GO nanoparticles whose lateral size differed by almost one order of magnitude: 100–200 nm and 1–5 μm. GO was modified with linear and eight-armed branched PEG. We grafted these two types of PEG polymer to GO based on results obtained by Xu et al. [19] who demonstrated that branched PEG provides better colloidal stability to GO nanoparticles in comparison with linear PEG. Experiments were performed with monocytes isolated from human peripheral blood using magnetic beads conjugated with anti-CD14 antibodies.

The aim of the study was to assess the effect of PEGylated GO on the functions of monocytes that are relevant for pathogen elimination and polarization of immune response.

The goals of the current study included:Synthesis and characterization of PEG-modified GO nanoparticlesAssessment of protein adsorption on PEG-modified GO nanoparticlesStudying the effects of PEGylated GO on the viability of monocytes, nanoparticle uptake, cytokine production, phagocytosis, and ROS formation.

## 2. Materials and Methods

### 2.1. Materials and Instrumentation

Graphene oxide powder (lateral size of 100–200 nm and 1–5 μm) was obtained from Ossila Ltd. (Sheffield, UK). Monochloroacetic acid, 1-(3-dimethylaminopropyl)-3-ethylcarbodiimide hydrochloride (EDC), N-hydroxysuccinimide (NHS), 8-arm-PEG-NH_2_ (10,000 kDa), methoxy polyethylene glycol amine (5 kDa) were obtained from Alfa Aesar (Ward Hill, MA, USA). Dichlorodihydrofluorescein diacetate (DCFH), polymyxin B, bovine serum albumin (BSA), lipopolysaccharide (LPS) were obtained from Sigma-Aldrich (Burlington, MA, USA). RPMI-1640 culture media were obtained from Biolot (St Petersburg, Russia). Ficoll-verografin was obtained from Dia-M (Moscow, Russia). Anti-CD14 MACS^®^ MicroBeads and MS Columns were obtained from Miltenyi Biotec (Bergisch Gladbach, Germany). ELISA kits were from Vektor-Best (Novosibirsk, Russia).

Solutions were prepared with deionized water. All reagents were used without additional purification.

Instrumentation. IFS 66/S IR spectrometer and Raman microscope SENTERRA were obtained from Bruker Corporation (Billerica, MA, USA). UV 2600 two-beam spectrophotometer was obtained from Shimadzu (Kyoto, Japan). ZetaPALS was obtained from Brookhaven Instruments Corporation (Wakefield, MA, USA). TGA/DSC 1 combined with TG-DSC device was obtained from Mettler-Toledo (Greifensee, Switzerland). CytoFlex S flow cytometer was obtained from Beckman Coulter (Brea, CA, USA).

### 2.2. PEGylation of GO

Carbodiimide-mediated coupling of PEG-NH_2_ and GO was made as follows. Aqueous solutions of GO (2 mg/mL) were sonicated for 30 min using probe sonicator (output power was 25 W for small GO and 150 W for large GO), and then carboxylated by the addition of Cl-CH2-COOH (18.5 mM) under sonication for 60 min. Carboxylated GO nanoparticles were washed by centrifugation at 10,000 g with water until neutral pH level was reached. EDC (to 4 mM), NHS (to 10 mM), and branched or linear PEG (to 2 mg/mL) were added to the suspension of carboxylated GO (pH 5.6) over the course of 5 min with constant ultrasonic treatment. The suspension was kept for 24 h at room temperature, GO-PEG was purified by dialysis, washed three times with ethanol by centrifugation (10,000 g), and dried under vacuum at +65 °C.

Large (1–5 μm) GO coated with linear and branched PEG will be further referred to as LnGO and LbGO, respectively. Small (100–200 nm) GO coated with linear and branched PEG will be further referred to as SnGO and SbGO, respectively. Designation “GO-PEG” will be used as a generic term for any type of PEGylated graphene oxide.

### 2.3. Characterization of GO and GO-PEG

FTIR spectra of the intact and PEGylated GO were obtained in the range 400–4000 cm^−1^ using KBr tablets (2 mg of nanoparticles per 299 mg of KBr). UV-Vis spectra of GO-PEG were recorded in the range of 200–900 nm. Hydrodynamic diameter and zeta potential were measured by dynamic light scattering (DLS). The percentage of the attached polymer was quantified by thermogravimetric analysis (TGA) at a heating rate of 10 K min^−1^ from 30 to 900 °C in an inert atmosphere. Raman spectra (D and G bands intensities) were obtained in the range 500–2000 cm^−1^ (laser power: 0.2 mW; wavelength: 532 nm). Scanning electron microscopy (SEM) images were obtained under an acceleration voltage of 20 kV.

### 2.4. Isolation of Monocytes from Human Blood

Venous blood was collected from the donors (healthy subjects, n = 7, 25–35 years old) by venipuncture with vacuum tubes. Written informed consent was obtained from all subjects who participated in the study. Peripheral blood mononuclear cells (PBMCs) were isolated from the venous blood by ficoll-verografin (1.077 g/L) gradient centrifugation. Heparinized blood was 1:1 mixed with RPMI-1640, and then 6 mL of the resulting mixture were layered upon 3 mL ficoll-verografin solution. After centrifugation for 40 min at 400 g (+25 °C) PBMC located above the ficoll-verografin layer were collected, diluted 5–10 times with RPMI-1640, and washed three times with RPMI-1640 (2× at 350 g for 20 min and 1× at 200 g for 20 min). The cell pellet was diluted in 1 mL of RPMI-1640. The concentration of PBMC was measured using a Neubauer chamber.

Monocytes (CD14-positive cells) were purified from PBMC fraction by positive immunomagnetic separation using magnetic microbeads carrying anti-CD14 antibodies according to the manufacturer’s instruction. Isolated monocytes were diluted to the required concentrations in Hanks’ balanced salt solution (HBSS).

### 2.5. Protein Adsorption Study

GO-PEG (100 μg) were dispersed in RPMI-1640 with 10% pooled human serum, or 10% fetal calf serum, or 0.7% BSA, which is equivalent to the total protein concentration in 10% human serum. The resulting concentration of GO-PEG was 5 and 25 μg/mL, therefore the volume of nanoparticle suspension was 20 and 4 mL respectively. Nanoparticles were kept at +37 °C for 60 min on the rotator, then washed by centrifugation (10,000 g, 30 min, three times) with phosphate buffer, re-dispersed in 20 μL of phosphate buffer, heated at +95 °C for 5 min, and analyzed by gel-electrophoresis in 10% polyacrylamide gel (SDS-PAGE). Gels were prepared using 0.1 M TRIS-HCl buffer, pH 8.8 with 0.1% SDS. Electrode buffer was TRIS-glycine buffer, pH ~ 8.4. Concentrating gel was not used. Electrophoresis was performed at a constant voltage of 200 V. Gels were stained with Coomassie G-250 diluted in a fixing solution (20% methanol, 7.5% acetic acid) and destained in a fixing solution. Destained gels were photographed using a smartphone. Photographs were cropped and presented without any color manipulation. Amount of protein in each sample was semiquantitatively measured with the aid of Image J software. For this, “Analyze > Measure” tool was used, which allows measurement of the mean grey value parameter, which characterizes mean brightness of each lane (Appendix A). The more protein the lower brightness. Mean grey value is expressed in arbitrary units and varies from 0 (minimal brightness) to 255 (maximal brightness). Detailed information can be found at https://imagej.nih.gov/ij/ (accessed on 28 November 2021).

### 2.6. Uptake of GO-PEG by Monocytes

Monocytes were diluted in 10% pooled human serum in RPMI-1640 to 2 × 10^6^ cells/mL. Fifty microliters of the cell suspension were combined with an equal volume of GO-PEG (resulting concentration of GO-PEG was 5 or 25 μg/mL). Samples were incubated at +4 or at +37 °C for 30 min and then on ice for 5 min. Cells were washed (350 g, 7 min) with a cold phosphate buffer and kept on ice until analysis by flow cytometry. The uptake of GO-PEG was assessed by measuring fluorescence in the PE-Cy7 channel (λex = 488 nm; bandpass filter: 720–840 nm). The percentage of GO-PEG-positive cells in the monocyte gate was determined (Figure 1). The phagocytosis index was determined by dividing the geometric mean of the fluorescence of the cells in the Cy7-positive gate (a linear gate placed to the right of the peak of cells in the sample without GO-PEG) by the number of cells in this gate.

### 2.7. Cell Viability

Monocytes were incubated with GO-PEG for 24 h; then they were washed and mixed 1:1 with 0.4% Trypan Blue. The percentage of stained (dead) cells was measured by visual assessment using a light microscope (100 cells were counted per sample, and 2 replicates were taken).

### 2.8. Phagocytosis of Escherichia coli by Monocytes

Monocytes (2 × 10^6^ cells/mL) in Phenol Red-free RPMI-1640 supplemented with 10% fetal calf serum were mixed with GO-PEG (5 and 25 μg/mL) and incubated for 30 min or 24 h in CO2 incubator (5% CO_2_; +37 °C). Then, an equal volume of FITC-labelled *E. coli* K12 (laboratory strain; 2 × 10^7^ cells/mL) was added, and the mixture was incubated for another 30 min. After that, cells were placed on ice for 10 min, washed once with 2 mL of cold buffer (phosphate buffer, pH 7.4 with 0.5% BSA and 0.1% sodium azide). The tubes were then placed on ice and analyzed on a CytoFlex S flow cytometer. The percentage of FITC-positive cells in the monocyte gate was determined in each sample. The phagocytosis index was determined by dividing the geometric mean of the fluorescence of the cells in the FITC-positive gate (a linear gate placed to the right of the peak of cells in the sample without *E. coli*) by the number of cells in this gate.

### 2.9. Production of Reactive Oxygen Species (ROS) by Monocytes

Luminol and lucigenin chemiluminescence. Hanks’ balanced salt solution, monocytes (to 10^5^ cells/mL), GO-PEG (to 5 or 25 μg/mL), and pooled heat-inactivated human serum (to 10%) were added to the wells of 96-well plate. Lucigenin or luminol was then added to 20 μM and 5 μM, respectively. Finally, opsonized zymosan was added to 1.5 μg/mL. Measurements of luminescence were performed for 90 (luminol) or 60 min (lucigenin) at +37 °C with 3 min intervals. Results are expressed in relative luminescence units (RLU) as the area under luminescence vs time curve (AUC) obtained by summarizing luminescence intensities at each time point.

DCFH fluorescence. Monocytes were stained with 100 μM DCFH for 30 min at +37 °C in a dark chamber; then, cells were washed with Hanks’ balanced salt solution. Monocytes concentration was adjusted to 2 × 10^6^ cells/mL. Fifty microliters of the cell suspension were transferred to a 96-well plate. After that, Hanks’ balanced salt solution, GO-PEG (to 5 or 25 μg/mL), and pooled human blood serum (to 10%) were added. Opsonized zymosan was added to 1.5 μg/mL. DCFH fluorescence was measured for 60 min at +37 °C with 3 min intervals. Results are expressed in relative fluorescence units (RFU) as the area under fluorescence vs time curve (AUC) obtained by summarizing luminescence intensities at each time point.

### 2.10. Cytokine Production

Monocytes (100 μL, 2 × 10^6^ cells/mL) were added to round-bottom 96-well plates, then RPMI-1640, pooled heat-inactivated human serum (final concentration of 10%), polymyxin B (10 μM) [20], and GO-PEG (5 or 25 μL) were added. Control samples without GO-PEG or containing LPS, 100 ng/mL) were also prepared. Cells were grown at 5% CO_2_ at +37 °C for 24 h. Then, plates were centrifuged (10 min, 400 g, + 25 °C) on a plate rotor, culture fluid was collected, and centrifuged at 20,000 g for 20 min at +4 °C. Supernatants were stored at −20 °C. Cytokine measurements were made by ELISA.

### 2.11. Statistical Analysis

Statistical analysis was performed in GraphPad Prism 6.0. Friedman’s test or one-way repeated measurements ANOVA with appropriate post-hoc tests were used for group comparison. In ROS production experiments, two technical replicates were completed for each donor. Average values of these replicates were used in further analyses. In other experiments, there were no technical replicates.

## 3. Results and Discussion

### 3.1. Synthesis and Characterization of GO-PEG

Effects of GO nanoparticles on living systems depend on many factors including lateral size, thickness, type of coating, elemental composition, and so on [21]. GO nanoparticles with lateral sizes of 100–200 nm and 1–5 μm were functionalized with PEG, a polymer capable of improving colloidal stability and the toxicity profile of nanomaterials. Two types of PEG were used for nanoparticle coating, namely linear PEG and branched PEG (8-armed PEG), which is based on literature data suggesting that the branched form may provide better stability in cell culture fluid [19]. Besides, previous studies have demonstrated that linear PEG is more intensively uptaken by phagocytes [22,23], to which monocytes belong, indicating that the results of interaction between cells and GO coated with linear and branched PEG can also be different.

Large (1–5 μm) GO coated with linear and branched PEG will be further referred to as LnGO and LbGO, respectively. Small (100–200 nm) GO coated with linear and branched PEG will be further referred to as SnGO and SbGO, respectively. Designation “GO-PEG” will be used as a generic term for any type of PEGylated graphene oxide.

Pristine GO nanoplatelets were treated with chloroacetic acid to introduce additional carboxylic groups. Then, aminated PEG was covalently attached to carboxylated GO via carbodiimide chemistry. The resulting nanoparticles were characterized by various techniques including FTIR, Raman spectroscopy, elemental analysis (EDS), TGA, SEM, and DLS (Figure 2).

#### 3.1.1. FTIR

Typical intense absorption bands (a.b.) at 3400 cm^−1^ (-OH), 1720 cm^−1^ (C=O), 1600 cm^−1^ (C=C), 1220 cm^−1^ (C-O), and 1070 cm^−1^ (C-O-C) were observed in the IR spectrum of the pristine graphene oxide, evidencing the presence of carbonyl, carboxyl, alkoxy, epoxy and hydroxyl groups in the sample. Spectra were independent of graphene oxide size. Following the pegylation one could observe both changes in the intensity of the pre-existing absorption bands and the emergence of novel typical absorption bands. Specifically, a.b. at 2870 cm^−1^ (-CH_2_-) and 1640 cm^−1^ (-NH-CO-) are detected and the a.b. intensity at 1070 cm^−1^ (C-O-C) is significantly elevated, indicating the presence of PEG in the system as well as for the amide bond formation between amine groups of PEG and GO. One should note the weakening in the a.b. intensity at 1420 cm^−1^ (C-OH) that additionally supports the amide bond formation between -COOH groups of GO and -NH_2_ groups of PEG-NH_2_.

#### 3.1.2. Raman Spectroscopy

Using Raman scattering spectroscopy it was found that GO PEGylation was manifested in an increase in oxygen amount (I_D_/I_G_ increases from 0.93 to 0.98) and decrease in the proportion of carbon moieties with sp^2^ bonds. Spectra are independent of graphene oxide size. Values obtained on the relationships of typical band intensities (I_D_/I_G_) could be interpreted as showing that the processes of carboxylation and subsequent PEGylation did not destroy the aromatic structure of monolayered graphene oxide.

#### 3.1.3. DLS

Hydrodynamic diameters and polydispersity indices of GO-PEG are presented in Table 1. Size distribution of SbGO and SnGO is presented in Figure 2C. DLS technique is based on the Einstein–Stokes equation describing the behavior of spherical particles. Being 2D material, GO cannot be accurately characterized by DLS, however, in general, obtained results reflect the size difference between larger and smaller nanoparticles. The zeta potential of GO-PEG is lower than −30 mV, that facilitates their good colloidal stability upon storage in deionized water.

#### 3.1.4. TGA

It is known that GO loses weight in an inert atmosphere at 150–300 °C due to the thermal decomposition of oxygen-containing groups (Appendix A). This thermal decomposition is accompanied by an exothermic effect. For the studied samples, SbGO and LbGO in a narrow temperature range of 150–170 °C, sharp decreases in mass are observed (Figure 2F,G), which is probably associated with the transformation of the various oxygen-containing GO groups (carbonyl, carboxyl, alkoxy, epoxy, and hydroxyl groups). For PEGylated samples, one more stage of weight loss is observed in the temperature range 250–450 °C, caused by thermal decomposition of the main chains of the branched PEG polymer. Calculations based on TGA data showed that both SbGO and LbGO contain about 20 wt% of PEG (Table 1).

The results obtained correspond to those for the samples of GO coated with linear PEG, with the only difference being that the intensities of thermal effects differ and the weight drops in the first temperature range of 150–300 °C are not so pronounced. The character of the TGA dependences for non-coated graphene oxide is similar to the literature data. However, the data obtained indicate a high oxidation state of the small GO nanoparticles (about 58 wt% of oxygen). Calculations based on TGA data showed that the SnGo and SnGO samples contain about 17–19 wt% of PEG (Table 1).

#### 3.1.5. SEM and EDS

Figure 2 shows typical scanning electron micrographs of the intact GO and LbGO. The rougher surface of PEGylated GO indicates the presence of polymer.

The presence of the PEG polymer on the GO surface was confirmed by elemental analysis using energy-dispersive X-ray spectroscopy (EDAX), implemented on a scanning electron microscope. Elemental analysis was performed on the intact GO and SbGO. Examples of mapping and spectra are shown in Appendix A. Intact GO is characterized by the presence of only carbon and oxygen in a ratio of 85:15 at%. In the SbGO sample, in addition to carbon and oxygen, the presence of nitrogen was observed. The atomic ratio of elements in the sample was changed to 74:2:24 (C:O:N). These changes are most likely also associated with the appearance of the branched PEG on the surface of GO nanoparticles.

### 3.2. Protein Adsorption by the PEGylated Graphene Oxide

When encountering cultural fluid containing 10% human serum GO-PEG nanoparticles accumulate serum proteins on their surface [24]. This so-called protein corona can sufficiently impact the interaction with monocytes. Protein corona of GO-PEG nanoparticles was studied by SPS-PAGE. One hundred μg of GO-PEG was incubated in a culture medium containing 10% human serum in concentrations of 5 and 25 μg/mL. Nanoparticles were collected and washed, then adsorbed proteins were eluted and analyzed by SDS-PAGE. Amount of adsorbed protein was semi-quantitatively measured by the assessment of total protein band intensity (brightness of corresponding lanes on the stained gel). Decrease of brightness indicated higher amount of adsorbed protein.

As expected, small nanoparticles adsorbed more protein than larger ones due to larger specific surface area (Figure 3). Nevertheless, PEGylation can not completely diminish protein adsorption on GO nanoparticles. Branched PEG more efficiently shield nanoparticle surfaces, which can be explained by steric hindrance generated by multiple PEG chains. These data are in agreement with previous studies reporting less intense cell attachment [25] and lower protein absorbance [26] for nanoparticles coated with branched PEG. Unexpectedly, we observed that the adsorption of serum proteins positively correlated with the concentration of nanoparticles. Considering that the concentration of protein in 10% serum (6–8 mg/mL) is 100–1000-fold higher than that of GO-PEG, one could expect that the surface of nanoparticles will be saturated with protein molecules regardless of nanoparticle concentration. Even if saturation of the GO-PEG surface was not reached at mentioned protein concentration, nanoparticles at 5 μL should adsorb more protein because of the higher protein-to-nanoparticle mass ratio. Therefore, an additional experiment was performed. We substituted 10% human serum with 7 mg/mL BSA. When incubating with BSA, GO-PEG adsorbs the same quantity of protein independent of their concentration (Appendix A). Therefore, it can be inferred that blood serum contains some factors facilitating higher protein adsorption at larger GO-PEG concentrations. Extraordinary interaction of GO with serum proteins has already been demonstrated by Kenry et al. [27]. The authors studied the adsorption of albumin, fibrinogen, and IgG on unmodified GO nanoparticles with various lateral sizes. For BSA a typical relationship was observed: the smaller the nanoparticles the lower adsorption, while contrarily, for fibrinogen adsorption, the pattern was opposite. Moreover, for IgG, no clear dependence of adsorption on lateral size was observed. Besides, at a certain concentration of IgG and fibrinogen, their adsorption sharply increased by about an order of magnitude in comparison with 2.5-fold lower concentrations.

A possible explanation of our findings is that higher concentration nanoparticles interact with unidentified serum proteins leading to their multi-layered adsorption. Probably, the formation of complexes, containing several nanoparticles and multiple proteins takes place, because the increase of GO-PEG concentration increases protein adsorption. Phenomena of multi-layered adsorption [28], as well as the dependence of protein adsorption on nanoparticle concentration [29], have been reported for other nanomaterials.

### 3.3. Uptake and Cytotoxicity of GO-PEG in Monocytes

The uptake of nanoparticles is an important characteristic of their interaction with immune cells. Monocytes being professional phagocytic cells are able to ingest particles from surrounding media. We studied whether the coating type and size of GO nanoparticles are responsible for uptake by human blood monocytes. Nanoparticles (5 and 25 μg/mL) were cultured with monocytes in the presence of 10% human blood serum for 30 min at +4 °C and +37 °C. Incubation at low temperature represents physical adsorption of GO-PEG whereas both adsorption and uptake of GO-PEG occurs at +37 °C allowing discrimination between adsorption and phagocytosis of nanoparticles [30,31]. The percentage of GO-PEG-positive monocytes and phagocytosis index (amount of GO-PEG nanoparticles per monocyte) were assessed with flow cytometry by measuring the fluorescence of graphene nanoparticles in Cy7 channel [32].

The amount of SbGO and LbGO per one monocyte was lower in comparison with SnGO and LnGO when nanoparticles (25 μg/mL) were incubated with cells at +37 °C (Figure 4D, *p* < 0.001). It was only at +4 °C that a difference for LnGO vs LbGO (*p* = 0.004) but not for SbGO vs SnGO (*p* = 0.468) was observed (Figure 4C). However, no distinct relationship between the type of PEG (branched or linear) and the number of GO-PEG-positive monocytes was revealed (Figure 4A,B). As expected, the percentage of positive cells was larger at higher GO concentrations. We did not perform a direct comparison of small and large GO-PEG nanoparticles due to their significant size (and, hence, fluorescence) difference.

Branched PEG decreases the number of GO nanoparticles, uptaken by a single cell, however, the number of monocytes that engulfed nanoparticles does not depend on the type of PEG polymer. Our data coincides with findings reported by Vila et al. and Matesanz et al. [22,23]. They showed that branched PEG-modified GO was 2–6-fold less intensively uptaken by various cell lines (osteoblasts and macrophages) than GO coated with linear PEG. Protein adsorption on the nanoparticles (protein corona) can alter the phagocytosis of nanoparticles. Duan et al. reported lower uptake of GO nanoparticles in the presence of 10% blood serum than in serum-free culture medium [33]. Moreover, it has been shown that certain serum proteins contribute to nanoparticle internalization, in particular that human serum albumin promotes uptake whereas the presence of clusterin in protein corona decreases it [34]. Although we did not study the composition of the GO-PEG protein corona, the GO nanoparticles stabilized with branched PEG showed lower total protein adsorption (Section 3.2.). We suppose that better stealth properties of branched PEG can explain the less intense uptake of GO nanoparticles coated with this type of polymer.

A higher percentage of GO-PEG-positive monocytes at larger GO-PEG concentrations is probably due to the lower number of GO-PEG nanoparticles in cells. Some of the monocytes contain GO-PEG, however the fluorescence of these nanoparticles is not high enough to be detected by the flow cytometer.

The viability of monocytes was assessed by Trypan Blue staining. The percentage of live cells in control was relatively low (median; 1–99 percentiles: 73.0; 63.0–77.0), which is probably due to the isolation procedure. In general, GO-PEG did not alter the cell viability (*p* values > 0.999 vs control for all groups in the Friedman test) however we should note that median viability values were slightly lower when 25 μg/mL of GO-PEG was added (Figure 5). Besides, SnGO at 25 μg/mL decreased the viability of monocytes obtained from 3 of 4 analyzed individuals to 47–51%. Similarly, in previous studies with both cell lines and cells isolated from mice (peritoneal macrophages, splenocytes, and bone marrow-derived dendritic cells) GO-PEG had no [6,35] or negligible (3–10% decrease) [22,23,36,37] impact on cell viability. In agreement with the earlier report [23], we did not observe any difference between the cytotoxicity of GO modified with linear and branched PEG.

### 3.4. Phagocytosis

Monocytes realize defense functions against pathogens via numerous mechanisms including phagocytosis and intracellular killing. Previous works have reported that graphene-based and other nanomaterials can both enhance [38,39] and suppress [40,41] the ability of phagocytic cells to engulf pathogens.

We studied whether interaction with graphene oxide nanoparticles influences the phagocytosis capacity of monocytes. Monocytes were preincubated with graphene oxide for 30 min or 24 h at +37 °C, then FITC-labelled *E. coli* were added to graphene-treated monocytes and kept for 30 min at +37 °C. Percentage of monocytes that engulfed bacteria, as well as the number of bacteria per monocyte (phagocytosis index) were measured.

Pre-incubation of monocytes with GO-PEG for 30 min did not affect the *E. coli* uptake. Prolonged graphene pre-treatment increased the number of FITC-positive monocytes in the control sample, perhaps due to the restoration of phagocytosis ability that was impaired by the cell isolation process (Figure 6, control samples).

Monocytes treated with 25 μg/mL of GO-PEG for 24 h less intensively engulfed bacteria: percentage of *E. coli*-positive monocytes declined by 10–15% in comparison with control cells (monocytes that did not interact with GO-PEG). The most prominent decrease was observed in monocytes contacted with 25 μg/mL of SbGO (15% decrease; *p* = 0.010) and LbGO (14% decrease; *p* = 0.002) (Figure 6).

Phagocytosis index was also lower in monocytes incubated with 25 μg/mL of GO-PEG for 24 h, however statistical comparison with the control group is underpowered due to small sample size and large between-individuals variability.

The decline of phagocytosis percentage cannot be explained solely by reaching the limit of phagocytosis capacity by monocytes after GO-PEG pre-treatment, because several studies have reported enhanced phagocytic activity after interaction with nanomaterials [38,39]. The possible mechanism of inhibition of phagocytosis is an interaction of GO-PEG with F-actin [22,40]. This interaction affects the cytoskeleton and, hence, might change the phagocytic activity of monocytes.

### 3.5. ROS

Monocytes produce reactive oxygen species to destroy pathogens [42,43]. In addition, ROS can increase the production of pro-inflammatory cytokines by monocytes/macrophages [44,45], as well as inducing apoptosis of T cells and monocytes themselves [46,47]. The regulation of ROS production by monocytes depends on the balance between the activation of NADPH oxidase (a multicomponent enzyme system) and the cellular levels of various antioxidant molecules [48].

Disruption of this regulation may lead to excessive oxidative stress, which in turn may serve as one of the links in the pathogenesis of diseases such as Alzheimer’s disease, chronic obstructive pulmonary disease, atherosclerosis, cancer, and others [49].

The Graphene family of nanomaterials change ROS production in immune cells [50]. Moreover, stimulation of ROS production is considered as one of the key mechanisms of graphene cytotoxicity [51]. ROS formation was measured with fluorescent and chemiluminescent tracers: luminol, lucigenin, and DCFH. Monocytes were mixed with GO-PEG and tracer, then luminescence was measured for 60 min. Opsonized zymosan was added to some samples in order to study the effects of GO-PEG on ROS production by activated monocytes. GO-PEG decreased luminol- and lucigenin-mediated luminescence (Figure 7). The observed effect was concentration-dependent. In the non-activated cells, GO-PEG decreased luminol-mediated chemiluminescence but did not affect lucigenin- and DCFH-mediated chemiluminescence. Size and PEG structure did not influence the results.

Despite many papers reporting that GO induces ROS formation in immune cells [50], GO can also hinder ROS production when acting as a ROS scavenger due to its multiple sp^2^ carbon atoms [5]. Besides, GO is an efficient luminescent quencher [52] and is, therefore, able to interfere with ROS detection techniques [35]. This fact makes the interpretation of results quite complex because the observed decrease of ROS production at higher GO concentration can be explained simply by its quenching ability, but not by an effect on cell physiology. We studied the influence of GO-PEG on H_2_O_2_-induced luminescence of lucigenin and luminol in cell-free conditions and revealed that GO-PEG at 25 μg/mL quenched lucigenin luminescence, but enhanced luminescence of luminol (Appendix A). Therefore, even if GO-PEG stimulated the production of ROS, the lucigenin-dependent luminescence induced by ROS production could not surpass the quenching effect of nanoparticles. Taking into account the enhancing effect of GO-PEG on luminol-dependent chemiluminescence, we suggest that GO-PEG can realize either inactivation of ROS produced by monocytes (antioxidant effect) or direct suppression of ROS production. Lucigenin mostly detects superoxide ions, whereas DCFH and luminol have less specificity and are able to detect various ROS species [53]. GO was shown to inactivate hydroxyl radicals [4,54] and superoxide radicals, but not hydrogen peroxide [4]. Therefore, we suggest that the inactivation of hydroxyl and superoxide radicals by GO-PEG can explain the decrease of chemiluminescence.

There are many reports on stimulation of ROS production by pristine GO, whereas PEGylated GO shows ambiguous effects. Six-armed and linear PEG-modified GO increased ROS production in osteoblasts and macrophages (RAW-264.7) [22]. Reduced GO-PEG stimulated production of intracellular ROS in mice bone marrow-derived dendritic cells and bone marrow mesenchymal stem cells [55,56]. At the same time, two research groups have demonstrated that GO-PEG did not stimulate the production of intracellular ROS in human breast cancer cells (MCF-7) [35,57]. GO-PEG had almost no effect on the ROS production by human neutrophils [58]. Direct comparison of our results with these data is challenging because the authors used various concentrations of GO-PEG, different incubation times, and ROS measurement modes.

### 3.6. Cytokines

Cytokines are small proteins secreted by both immune and non-immune cells which orchestrate homing, activation, and proliferation of immune cells as well as immune response polarization. Knowledge about the ability of GO-PEG to alter cytokine production by monocytes is essential for the development of GO-based adjuvants and therapeutics. We cultivated monocytes in the presence of GO-PEG and measured the concentration of three cytokines-TNFα, IL-6, IL-10. It is known that TNFα and IL-6 realize the pro-inflammatory regulatory activity of cytokines, while IL-10 has anti-inflammatory activity.

Before experiments, we measured the concentration of endotoxin (LPS) in GO-PEG preparations [59]. Obtained values were compared with data on the effect of endotoxin on monocytes obtained by Schwarz et al. [60] taking into account the dilution factor. The authors of the mentioned work reported that endotoxin at a concentration of 0.02 EU/mL did not change the production of IL-1beta, IL-6, IL-8, and TNFα as well as expression levels of CD40, CD80, CD83, and CD86 by human monocytes. The concentration of endotoxin measured by the LAL test in all GO-PEG preparations was higher than 0.02 EU/mL (Appendix A), therefore polymyxin B was added in cell culture medium to quench the effects of endotoxin. The amount of polymyxin B added was capable of inactivating endotoxin in concentration as high as 100 ng/mL being, therefore, a large excess in relation to measured endotoxin concentration. Excess of polymyxin B was used because LAL-test can underestimate endotoxin concentration due to its ability to adsorb on nanoparticles [61].

SnGO at 25 μL and LbGO at 5 and 25 μL increased the production of IL-10. For TNFα and IL-6 median cytokine concentrations were higher in comparison with control, but p values exceeded 0.1. Monocytes from four donors were tested in these experiments. The between-donor difference in both basal and GO-induced cytokine production was very high and reached an order of magnitude. Cells isolated from donors 1 and 2 had lower basal levels of cytokine secretion and were much less sensitive to GO-PEG compared with that of donors 3 and 4 (Figure 8). We suppose that there is some probability of SnGO and LbGO influence of TNFα and IL-6 production, however further studies are necessary. In general, there was no clear relation between the size of GO nanoparticles or the type of PEG, and the change of cytokine production. Despite polymyxin B being added, there is still a chance that endotoxin adsorbed on graphene surface and corona proteins avoided inactivation and interacted with LPS receptors after GO-PEG were uptaken by monocytes. This suggestion is based on results obtained by Jurgens et al. [62] who showed that human serum albumin binds multiple LPS molecules and decreases the binding of LPS to polymyxin B. Moreover, adsorption on nanoparticles can by itself significantly enhance the pro-inflammatory effect of LPS [63]. Orecchioni et al. reported that GO caused a non-cell-specific production of all analyzed cytokines in a variety of cell populations, whereas the effect of LPS was more specific [64].

Results of recent studies dedicated to the effects of GO-PEG on cytokine production by various immune cells are contradictory. Cytokine production depends on cell type [22] and grafting density of PEG molecules on GO [65]. We did not assess the number of PEG chains per one GO nanoparticle, but the mass fraction of PEG was almost the same in all GO-PEG. Some authors demonstrated enhanced secretion of TNFα by RAW-264.7 macrophages [36], peritoneal macrophages [65], and mice bone marrow-derived dendritic cells [56]. Interestingly, the production of TNFα by peritoneal macrophages decreased after 24 h of incubation but increased after 48 h [37]. In the same papers, the increase of TNFα production did not correlate with changes in the production of IL-10, IL-6, and other cytokines [36,56,65]. Importantly, most cited studies lack information about endotoxin content in tested nanoparticle preparations which significantly complicates the comparative analysis of their results.

## 4. Conclusions

Summarizing the obtained results, we can conclude that PEGylated graphene oxide influences the functions of monocytes such as cytokine production, phagocytosis, and ROS formation. The ability of GO-PEG to reduce phagocytosis of bacterial cells and ROS formation indicates that GO-PEG can potentially inhibit the elimination of pathogens by the immune cells. Graphene oxide is considered a promising component of antimicrobial therapeutics owing to its toxicity towards bacterial cells [66]. Inhibition of antibacterial properties of monocytes can, therefore, reduce the efficiency of GO towards pathogen elimination. An increase of cytokine production by the monocytes upon interaction with GO-PEG can change the polarization of immune response leading to inadequate and, hence, inefficient reaction of the immune cells on the pathogen, which is important when graphene is utilized as adjuvant in vaccines [7].

Endotoxin contamination of GO-PEG significantly complicates the interpretation and comparison of the results on immunotoxicity of GO-based nanomaterials. In this work, we used GO-PEG contaminated with endotoxin, so we had to add polymyxin B to the culture medium. Preparation of apyrogenic GO-PEG is a challenging task, which requires GO synthesized in endotoxin-free conditions. Such products are of limited commercial availability, whereas their in-house synthesis can be conducted only in a specialized laboratory. Therefore, optimized methods for the preparation of apyrogenic GO of desirable size are of great importance as well as methods for post-synthesis depyrogenation, which do not alter the structural integrity of GO-PEG.

## Figures and Tables

**Figure 1 nanomaterials-12-00126-f001:**
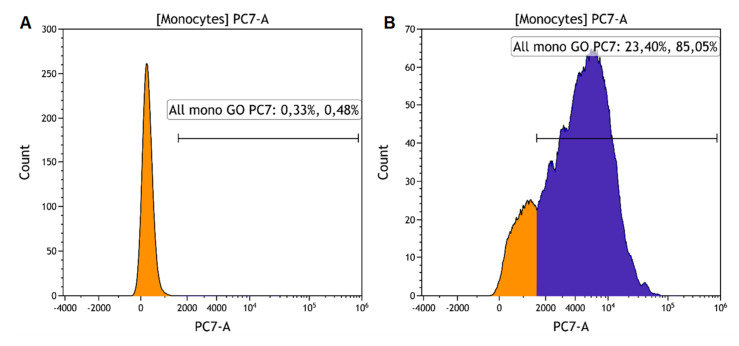
Histograms of cell fluorescence in PC-7 channel. (**A**)—control (cells without GO); (**B**)—sample with 25 μg/mL GO-PEG 1–5 μm. Note: Cells entering the gate of fluorescent (PC-7-positive) cells that absorbed/engulfed GO are highlighted in blue in the histogram.

**Figure 2 nanomaterials-12-00126-f002:**
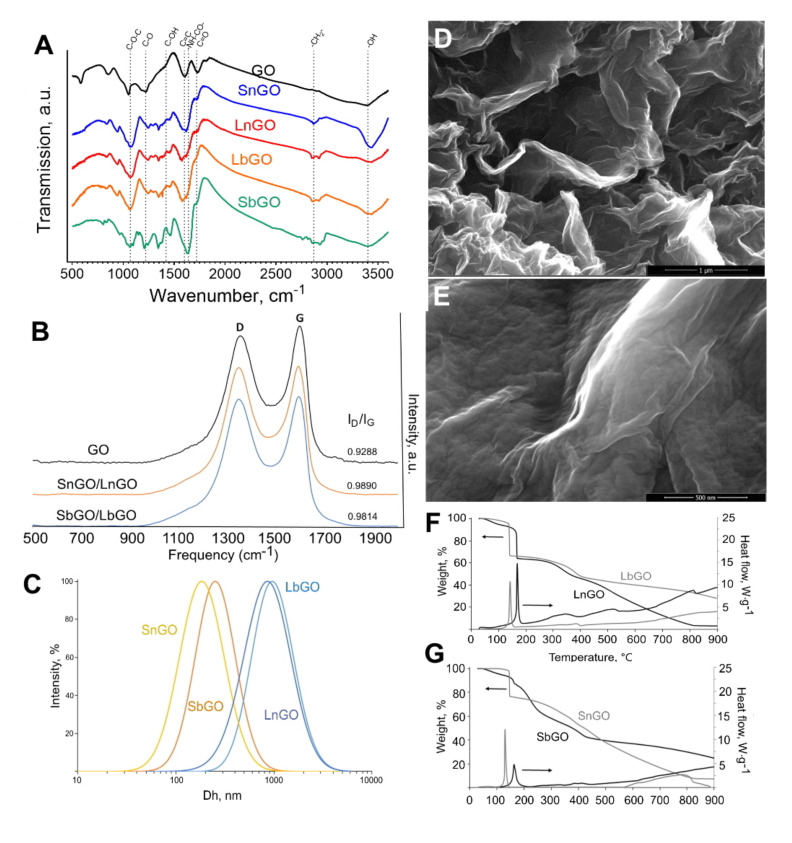
Characterization of GO-PEG. (**A**)—FTIR spectra; (**B**)—Raman spectra; (**C**)—intensity-weighted size distribution determined by DLS; (**D**,**E**)—SEM images of GO (**D**) and LbGO (**E**); (**F**,**G**)—TGA/DSC of GO-PEG. Scale bars are 1 μm (**D**) and 500 nm (**E**).

**Figure 3 nanomaterials-12-00126-f003:**
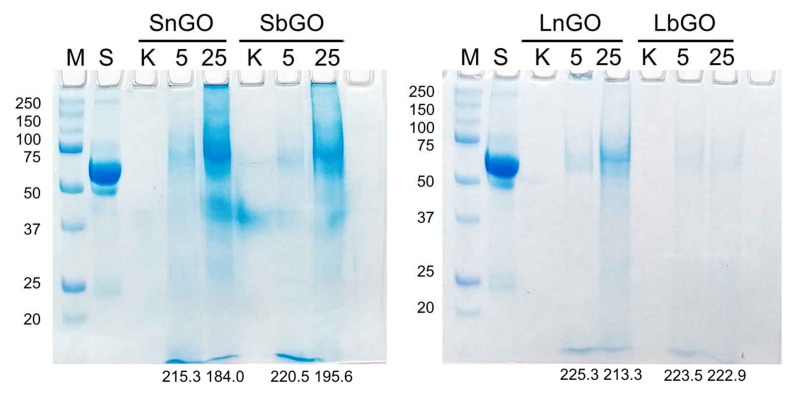
SDS-PAGE of GO-PEG protein corona. M—protein markers, their molecular weights are indicated on the left side. S-1% human serum. K—GO-PEG (25 μg/mL) incubated in culture medium without 10% human serum. GO-PEG (5 or 25 μg/mL) incubated in culture medium with 10% human serum. Mean gray values, which reflect amount of protein in sample, are presented below corresponding lanes.

**Figure 4 nanomaterials-12-00126-f004:**
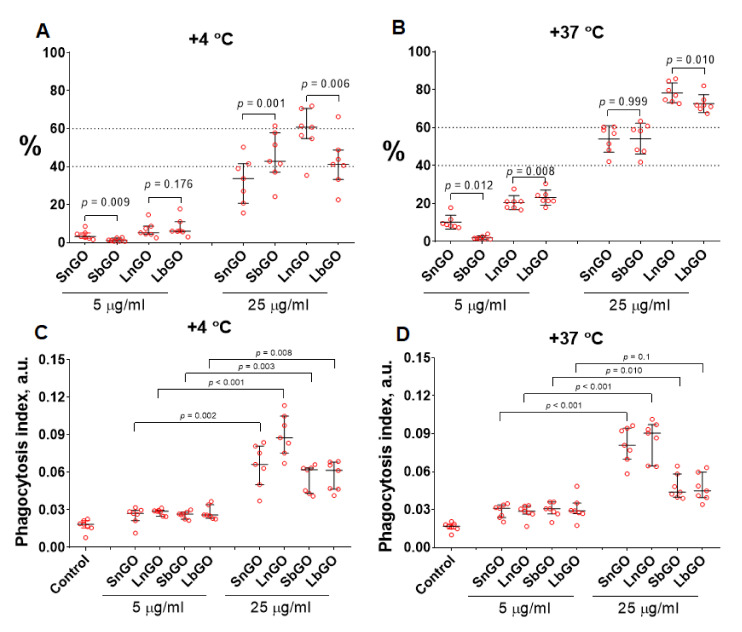
Percentage of GO-PEG positive cells in monocyte gate (**A**,**B**) and phagocytosis index (**C**,**D**). Incubation of monocytes with GO-PEG was carried out at +4 and at +37 °C. Dotted lines are drawn for easier visual comparison of results. n = 7, median and interquartile range are reported. Data were analyzed by repeated measurements one-way ANOVA with Sidak post-hoc correction for multiple comparisons.

**Figure 5 nanomaterials-12-00126-f005:**
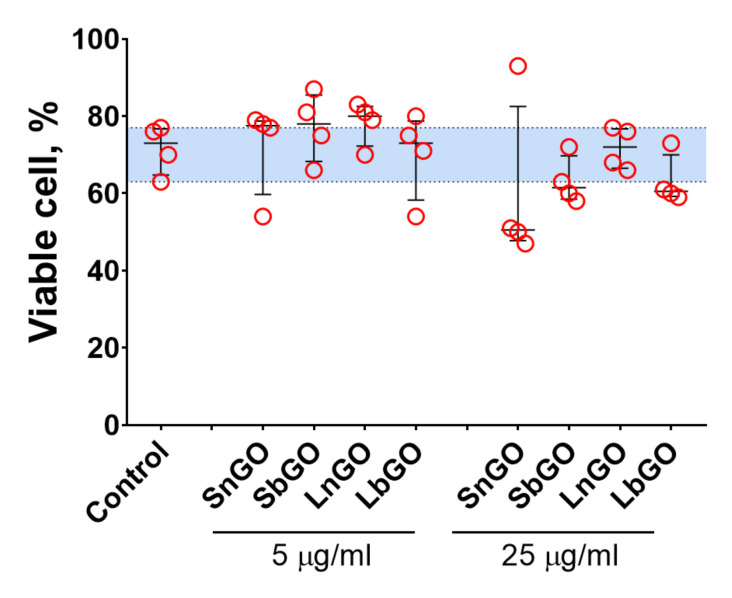
Viability of monocytes cultivated with GO-PEG for 24 h. n = 4, median and interquartile range are reported. Data were analyzed by Friedman test with Dunn’s post-hoc correction for multiple comparisons. Blue area represents 1–99 percentile range of control.

**Figure 6 nanomaterials-12-00126-f006:**
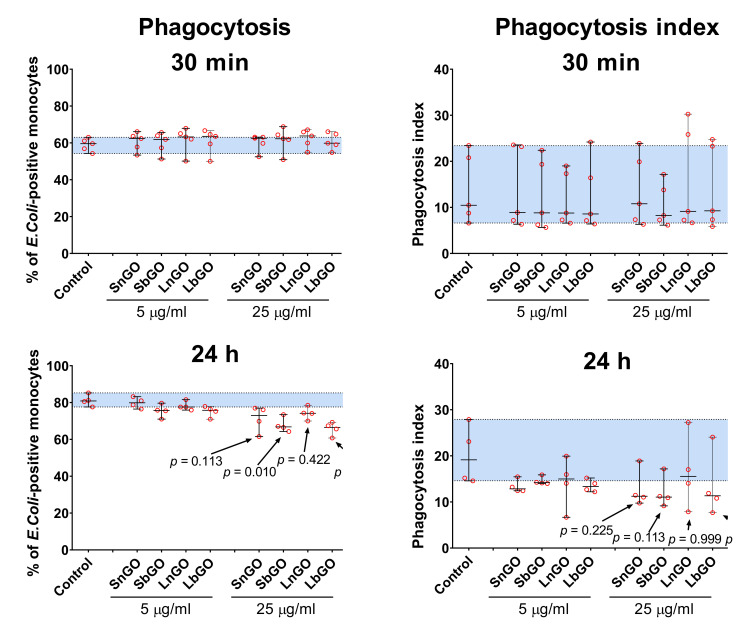
Phagocytosis of *E. coli* by monocytes and phagocytosis index. Monocytes were successively incubated with GO-PEG for 30 min or 24 h and with FITC-labelled *E. coli* for 30 min. The n = 5 (30 min) or n = 4 (24 h), median and interquartile range are reported. Data were analyzed by Friedman test with Dunn’s post-hoc correction for multiple comparisons. Blue area represents 1–99 percentile range of control. *p* values indicate comparison to control sample.

**Figure 7 nanomaterials-12-00126-f007:**
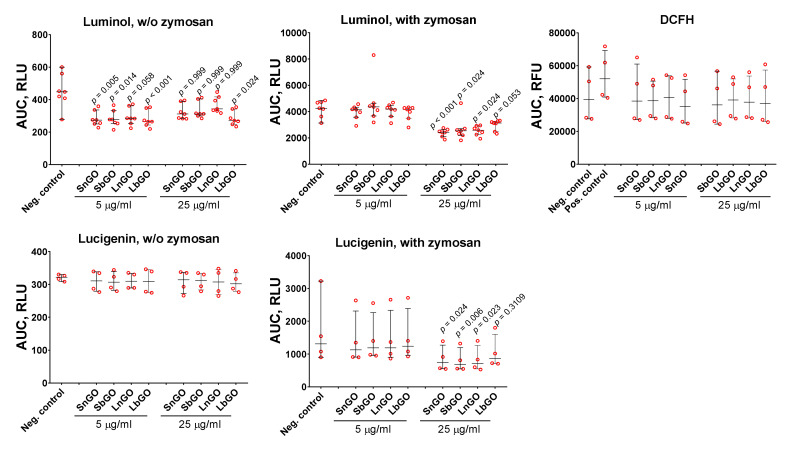
ROS formation by monocytes measured by luminescent tracers: luminol, lucigenin, and DCFH. In some samples zymosan was used as an activator. Negative control-cultural fluid without GO-PEG. Positive control-zymosan (DCFH experiment). n = 4 for DCFH and lucigenin experiments, n = 7 for luminol experiment, median and interquartile range are reported. Data were analyzed by Friedman test with Dunn’s post-hoc correction for multiple comparisons. AUC-area under luminescence/fluorescence vs. time curve. RLU-relative luminescent units. RFU-relative fluorescent units. *p* values indicate comparison with control sample.

**Figure 8 nanomaterials-12-00126-f008:**
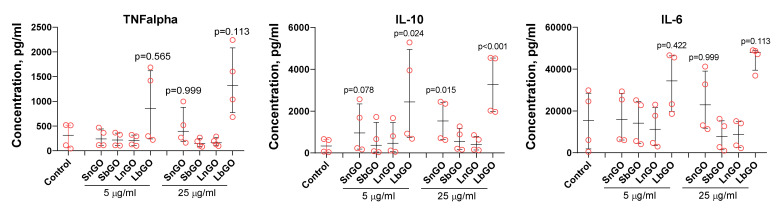
Cytokine production by monocytes cultivated with GO-PEG. Control—cultural fluid without GO-PEG. Polymyxin B (10 μM) was added in all samples. n = 4, median and interquartile range are reported. Data were analyzed by Friedman test with Dunn’s post-hoc correction for multiple comparisons. *p* values indicate comparison with control sample.

**Table 1 nanomaterials-12-00126-t001:** Properties of GO-PEG.

	SnGO	SbGO	LnGO	LbGO
Dh, nm ^1^	184 ± 73	287 ± 52	891 ± 18	1376 ± 48
PdI	0.25 ± 0.02	0.23 ± 0.02	0.21 ± 0.02	0.30 ± 0.01
Zeta Potential, mV	−31.70 ± 1.70	−34.28 ± 0.41	−39.98 ± 1.17	−53.56 ± 1.23
PEG Coverage, wt%	17.2 ± 1.4	20.5 ± 1.8	19.4 ± 2.2	20.5 ± 1.1

^1^ Dh—hydrodynamic diameter, PdI—polydispersity index.

## Data Availability

The datasets used and/or analyzed during the current study are available from the corresponding author on reasonable request.

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
