# Peer review of "Interaction of Graphene Oxide Modified with Linear and Branched PEG with Monocytes Isolated from Human Blood"

_nanomaterials, 2021, doi:10.3390/nano12010126_

Round 1

Reviewer 1 Report

The authors investigate the effect of graphene oxide nanoparticles on monocyte viability and function. The introduction is fairly well written, but much of the discussion is written in a confusing way. I indicate some specifics below. Specific comments are listed below.

  1. The naming convention of SnGO etc was a bit confusing at times. It was not always clear from the name whether the GOs were pegylated or not.

  1. It’s not clear to me in figure 4 how the authors can conclude that the percentage of cells is lower. It seems equally possible that they was just not sufficient numbers of particles at the lower concentrations taken up by the cells. Providing data that shows the intensity of the fluorescence in the lower concentration case versus the higher concentration would be useful.
  2. The entire first paragraph under section 4 Phagocytosis is awkwardly worded and hard to follow. It is difficult to figure out what the authors claim they observe versus what they are referencing from previous work.
  3. In figure 6, they annotate some of the panels with what appear to be p values, but there is no indication of what the comparison is that resulted in these p values. Is it between 5 and 25 u/ml or some other comparison? This seems to be true in several other figures as well.
  4. On line 415, I don’t understand what the authors are trying to say with the statement, “Monocytes, along with neutrophils, are the main producers of reactive oxygen spe- 415 cies, which are necessary for bacterial destruction”. Are they saying that monocytes produce the majority of ROS in the human body?
  5. The conclusion is a very vague summary.

Author Response

Dear Editor-in-Chief and Referees

We express our great gratitude to the Reviewer for comments and thoughtful suggestions. Based on these comments and suggestions, we have made modifications to the original manuscript. Additional analyses of data on cellular uptake of graphene oxide were performed. Conclusions we drew from these analyses differ from our initial conclusions. More specifically, we observed some influence of PEG structure on the uptake of PEGylated graphene oxide by monocytes.

Key changes we made are summarized below:

  1. Abstract was modified according to new findings.
  2. The aim of the study was clearly indicated.
  3. Additional section on statistical analyses (2.11) was added.
  4. Method of semi-quantitative assessment of adsorbed protein was added (Figure S1; Section 2.5.)
  5. Statistical analysis of uptake of PEGylated graphene oxide by monocytes was performed using other method (repeated measurements ANOVA). Phagocytosis indices were also calculated. Section 3.3. was re-written in accordance with obtained results and suggestions from the Reviewers.
  6. Section 3.4. and Conclusion were completely re-written.
  7. Figures were modified according to Reviewers’ comments.
  8. Three new references were added

All changes are highlighted with yellow.

We believe that the manuscript has been greatly improved and hope it has reached high journal's standard.

Sincerely yours

Pavel Khamtsov.

Reviewer #1

The authors investigate the effect of graphene oxide nanoparticles on monocyte viability and function. The introduction is fairly well written, but much of the discussion is written in a confusing way. I indicate some specifics below. Specific comments are listed below.

  1. The naming convention of SnGO etc was a bit confusing at times. It was not always clear from the name whether the GOs were pegylated or not.

Response

As we indicated in the Section 2.2. abbreviations like SnGO, SbGO etc. were used to designate various types of PEGylated graphene oxide. Non-PEGylated one was designated as GO. All abbreviations used in the text were defined in the Section 2.2. Based on suggestions from other Reviewers and comments from our colleagues we suppose that abbreviations allow clear identification of the nanoparticle type. For clarity, we have indicated the abbreviations ​​for the names of the graphene oxide samples at the beginning of the Results and Discussion.

  1. It’s not clear to me in figure 4 how the authors can conclude that the percentage of cells is lower. It seems equally possible that they was just not sufficient numbers of particles at the lower concentrations taken up by the cells. Providing data that shows the intensity of the fluorescence in the lower concentration case versus the higher concentration would be useful.

Response

We calculated phagocytosis index, which is proportional to the amount of GO-PEG nanoparticles per one monocyte using method described in Section 2.6. Indeed, the number of nanoparticles per cell was lower at a lower concentration of GO-PEG nanoparticles (results were added to Figure 4C,D). Moreover, we used more sensitive statistical method for group comparison (ANOVA) when analyzed GO-PEG uptake results. We found it appropriate to use parametric test (ANOVA) instead of non-parametric one (Friedman’s test) because distribution in samples was close to normal. Initially, we used the non-parametrical statistical methods throughout the study, because distribution of data in samples was unknown and samples were too small to perform normality test (e.g. Shapiro-Wilk test) with appropriate sensitivity. However, data points in GO-PEG uptake experiments had relatively homogeneous distribution with small between-individuals variability. Friedman’s test did not found differences even between samples which mean values were twice different. We decided, that application of Friedman’s test led to underreporting of findings. Therefore, we re-wrote the Section 3.3. in accordance with new analysis results and suggestions from the Reviewer.

  1. The entire first paragraph under section 4 Phagocytosis is awkwardly worded and hard to follow. It is difficult to figure out what the authors claim they observe versus what they are referencing from previous work.

Response

Section 3.4. Was completely re-written according to the Reviewer’s comments.

  1. In figure 6, they annotate some of the panels with what appear to be p values, but there is no indication of what the comparison is that resulted in these p values. Is it between 5 and 25 u/ml or some other comparison? This seems to be true in several other figures as well.

Response

Fixed

  1. On line 415, I don’t understand what the authors are trying to say with the statement, “Monocytes, along with neutrophils, are the main producers of reactive oxygen spe- 415 cies, which are necessary for bacterial destruction”. Are they saying that monocytes produce the majority of ROS in the human body?

Response

The sentence was changed

  1. The conclusion is a very vague summary.

Response

The conclusion was re-written

Reviewer 2 Report

Dear authors,
The manuscript is well written, easy to read and, in my opinion, with a logical sequence of the subjects.
Nevertheless, I have some minor comments/suggestions:
In introduction when you define graphene, it would be convenient to add that the carbon atoms present a sp2 bond hybridization. This clarification will allow readers, not so familiarized with this material, to be more aware that the hexagon is different from and hexane (just to give an example) were the ring of carbons present sp3 bond hybridization and hydrogens.
Considering the extent of the work that you developed, I was disappointed with the conclusions. They are very short, do not reflect your work and do not have any insight how your contributed (or can be compared) with the current state-of-art.
Moreover, if possible, I would like to read a little sentence/paragraph on your insights of the direction to follow to clarify the remaining doubts.

Author Response

Dear Editor-in-Chief and Referees

We express our great gratitude to the Reviewer for comments and thoughtful suggestions. Based on these comments and suggestions, we have made modifications to the original manuscript. Additional analyses of data on cellular uptake of graphene oxide were performed. Conclusions we drew from these analyses differ from our initial conclusions. More specifically, we observed some influence of PEG structure on the uptake of PEGylated graphene oxide by monocytes.

Key changes we made are summarized below:

  1. Abstract was modified according to new findings.
  2. The aim of the study was clearly indicated.
  3. Additional section on statistical analyses (2.11) was added.
  4. Method of semi-quantitative assessment of adsorbed protein was added (Figure S1; Section 2.5.)
  5. Statistical analysis of uptake of PEGylated graphene oxide by monocytes was performed using other method (repeated measurements ANOVA). Phagocytosis indices were also calculated. Section 3.3. was re-written in accordance with obtained results and suggestions from the Reviewers.
  6. Section 3.4. and Conclusion were completely re-written.
  7. Figures were modified according to Reviewers’ comments.
  8. Three new references were added

All changes are highlighted with yellow.

We believe that the manuscript has been greatly improved and hope it has reached high journal's standard.

Sincerely yours

Pavel Khamtsov

Reviewer #2

Dear authors,

The manuscript is well written, easy to read and, in my opinion, with a logical sequence of the subjects.

Nevertheless, I have some minor comments/suggestions:

In introduction when you define graphene, it would be convenient to add that the carbon atoms present a sp2 bond hybridization. This clarification will allow readers, not so familiarized with this material, to be more aware that the hexagon is different from and hexane (just to give an example) were the ring of carbons present sp3 bond hybridization and hydrogens.

Considering the extent of the work that you developed, I was disappointed with the conclusions. They are very short, do not reflect your work and do not have any insight how your contributed (or can be compared) with the current state-of-art.

Moreover, if possible, I would like to read a little sentence/paragraph on your insights of the direction to follow to clarify the remaining doubts.

  1. In introduction when you define graphene, it would be convenient to add that the carbon atoms present a sp2 bond hybridization. This clarification will allow readers, not so familiarized with this material, to be more aware that the hexagon is different from and hexane (just to give an example) were the ring of carbons present sp3 bond hybridization and hydrogens.

Response

The sentence was changed according to Reviewer’s comments.

  1. Considering the extent of the work that you developed, I was disappointed with the conclusions. They are very short, do not reflect your work and do not have any insight how your contributed (or can be compared) with the current state-of-art.

Response

Conclusion was re-written in accordance with Reviewer’s comments

  1. Moreover, if possible, I would like to read a little sentence/paragraph on your insights of the direction to follow to clarify the remaining doubts.

Response

In our opinion, one of the main obstacles in studying the effects of graphene-based nanomaterials on cell physiology is a limited availability of apyrogenic graphene preparations. Despite there are various methods to control lateral size of graphene oxide nanoparticles or to attach desirable polymer molecules, endotoxin contamination is still a great problem, which significantly complicates interpretation of results. We added short discussion on this point to the Conclusion.

Reviewer 3 Report

Khramtsov et al. studied the effects of different pegylated graphene oxides on the viability and cellular functions of monocytes. The study provides insight into potential impacts of GO-PEG on human immune cells if these nanomaterials would be used as nanomedicines. The nanomaterials were coated with linear or branched PEG and thoroughly characterized. Overall, the study results have been well presented and discussed, and the conclusions are supported by the data. It is a well-conducted study and the manuscript has been written clearly. There are minor issues in the method descriptions which could be more detailed. There are also a few typographical errors and some sentences would benefit from rephrasing. My specific comments are below. The references in the text are in an unusual format in that the author names are not followed by "et al." in the cases where there is more than one author. This will not be a problem after formatting the references to the style of the journal Nanomaterials which is a number style. Also, the reference list is currently both numbered and in alphabetical order which is unconventional. In the end of the Introduction, the authors have listed the specific goals of the study. However, the general aim of the study has not been stated. It would be helpful to also articulate the overall scientific goal of the study in the end of the Introduction. Materials and Methods Line 87, it could be stated if the graphene oxide materials were purchased as powders or suspensions. Line 92, “Sigma” should be “Sigma-Aldrich”. It could be added that RPMI-1640 is cell culture medium. Line 106, it could be specified if probe or bath sonication was used. Line 107, the volume and concentration of Cl-CH2-COOH should be specified. Line 108, please add how the washing was done. In section 2.3. “Characterization of GO and GO-PEG”, in the cases where the characterization was done in suspensions, it should be specified which medium or water was used and what was the nanomaterial concentration. In section 2.4. “Isolation of monocytes from human blood”, the source and circumstances of obtaining human blood for the study should be explained. Later in the manuscript 4 subjects are mentioned. This should be explained in the methods. Who were the individuals, were they healthy etc. Line 141, “CO-PEG” should be “GO-PEG” Line 147, the method of SDS-PAGE should be explained. For example, how was the gel stained, were the protein band sizes quantified somehow (later the authors compare the amounts of proteins associated with different GO-PEG but quantitative analysis has not been described). In section 2.6. “Uptake of GO-PEG by monocytes” it is mentioned that fluorescence was measured but it is unclear what was the source of fluorescence. Were the cells stained with a fluorescence stain? In the results it is mentioned that also side scatter was measured but this has not been described in the methods. Lines 164-165, specify how many cells were counted per sample and in how many replicates to obtain quantitative data by microscopy. Line 172, it is unclear, based on the composition of the staining buffer, what the stain was. Line 201, LPS has not been explained. A separate subsection about the statistical analysis of the data, replicates etc. should be added to the Materials and Methods section. Lines 207-208, add references to the sentence to support this claim. In the caption of Figure 2 and line 228, IR should be FTIR. The FTIR absorption bands and functional groups discussed in lines 227-237 should be indicated in the spectra in Figure 2. Otherwise, it is difficult to understand where these specific bands are because some regions contain multiple bands close together. Lines 264 and 271, “Table” should be “Table 1”. Lines 267-268, the meaning of the sentence is unclear, please rephrase. Lines 268-269, has the TGA data for pristine GO presented? I did not notice this. Lines 285-286, please add references to support the claim. Line 289, “human serum in concentrations of 5 and 25 µl”. This sentence needs to be corrected. Lines 297-298, “Branched PEG more efficiently shield nanoparticle surfaces…” This is not obvious from the comparison of SnGO and SbGO sample protein bands in Figure 3. Was any quantitative assessment done to compare the band sizes? Line 362, “HSA” has not been explained. Line 365, “the amount of adsorbed did not…” There appears to be a word missing here. Line 373, the meaning of “linear cells” is unclear. Figure 6, please correct “E. Coli” in the y-axis titles. Line 427, please add references to support the claim. Figure 7, please explain “AUC”, “RFU” and ”RLU” in the figure caption. Line 517, “GO caused a broad, non-cell-specific triggering the production of all cytokines”. There seems to be a word missing here.

Author Response

Dear Editor-in-Chief and Referees

We express our great gratitude to the Reviewer for comments and thoughtful suggestions. Based on these comments and suggestions, we have made modifications to the original manuscript. Additional analyses of data on cellular uptake of graphene oxide were performed. Conclusions we drew from these analyses differ from our initial conclusions. More specifically, we observed some influence of PEG structure on the uptake of PEGylated graphene oxide by monocytes.

Key changes we made are summarized below:

  1. Abstract was modified according to new findings.
  2. The aim of the study was clearly indicated.
  3. Additional section on statistical analyses (2.11) was added.
  4. Method of semi-quantitative assessment of adsorbed protein was added (Figure S1; Section 2.5.)
  5. Statistical analysis of uptake of PEGylated graphene oxide by monocytes was performed using other method (repeated measurements ANOVA). Phagocytosis indices were also calculated. Section 3.3. was re-written in accordance with obtained results and suggestions from the Reviewers.
  6. Section 3.4. and Conclusion were completely re-written.
  7. Figures were modified according to Reviewers’ comments.
  8. Three new references were added

All changes are highlighted with yellow.

We believe that the manuscript has been greatly improved and hope it has reached high journal's standard.

Sincerely yours

Pavel Khamtsov

Reviewer #3

Khramtsov et al. studied the effects of different pegylated graphene oxides on the viability and cellular functions of monocytes. The study provides insight into potential impacts of GO-PEG on human immune cells if these nanomaterials would be used as nanomedicines. The nanomaterials were coated with linear or branched PEG and thoroughly characterized. Overall, the study results have been well presented and discussed, and the conclusions are supported by the data. It is a well-conducted study and the manuscript has been written clearly. There are minor issues in the method descriptions which could be more detailed. There are also a few typographical errors and some sentences would benefit from rephrasing. My specific comments are below. The references in the text are in an unusual format in that the author names are not followed by "et al." in the cases where there is more than one author. This will not be a problem after formatting the references to the style of the journal Nanomaterials which is a number style. Also, the reference list is currently both numbered and in alphabetical order which is unconventional. In the end of the Introduction, the authors have listed the specific goals of the study. However, the general aim of the study has not been stated. It would be helpful to also articulate the overall scientific goal of the study in the end of the Introduction. Materials and Methods Line 87, it could be stated if the graphene oxide materials were purchased as powders or suspensions. Line 92, “Sigma” should be “Sigma-Aldrich”. It could be added that RPMI-1640 is cell culture medium. Line 106, it could be specified if probe or bath sonication was used. Line 107, the volume and concentration of Cl-CH2-COOH should be specified. Line 108, please add how the washing was done. In section 2.3. “Characterization of GO and GO-PEG”, in the cases where the characterization was done in suspensions, it should be specified which medium or water was used and what was the nanomaterial concentration. In section 2.4. “Isolation of monocytes from human blood”, the source and circumstances of obtaining human blood for the study should be explained. Later in the manuscript 4 subjects are mentioned. This should be explained in the methods. Who were the individuals, were they healthy etc. Line 141, “CO-PEG” should be “GO-PEG” Line 147, the method of SDS-PAGE should be explained. For example, how was the gel stained, were the protein band sizes quantified somehow (later the authors compare the amounts of proteins associated with different GO-PEG but quantitative analysis has not been described). In section 2.6. “Uptake of GO-PEG by monocytes” it is mentioned that fluorescence was measured but it is unclear what was the source of fluorescence. Were the cells stained with a fluorescence stain? In the results it is mentioned that also side scatter was measured but this has not been described in the methods. Lines 164-165, specify how many cells were counted per sample and in how many replicates to obtain quantitative data by microscopy. Line 172, it is unclear, based on the composition of the staining buffer, what the stain was. Line 201, LPS has not been explained. A separate subsection about the statistical analysis of the data, replicates etc. should be added to the Materials and Methods section. Lines 207-208, add references to the sentence to support this claim. In the caption of Figure 2 and line 228, IR should be FTIR. The FTIR absorption bands and functional groups discussed in lines 227-237 should be indicated in the spectra in Figure 2. Otherwise, it is difficult to understand where these specific bands are because some regions contain multiple bands close together. Lines 264 and 271, “Table” should be “Table 1”. Lines 267-268, the meaning of the sentence is unclear, please rephrase. Lines 268-269, has the TGA data for pristine GO presented? I did not notice this. Lines 285-286, please add references to support the claim. Line 289, “human serum in concentrations of 5 and 25 µl”. This sentence needs to be corrected. Lines 297-298, “Branched PEG more efficiently shield nanoparticle surfaces…” This is not obvious from the comparison of SnGO and SbGO sample protein bands in Figure 3. Was any quantitative assessment done to compare the band sizes? Line 362, “HSA” has not been explained. Line 365, “the amount of adsorbed did not…” There appears to be a word missing here. Line 373, the meaning of “linear cells” is unclear. Figure 6, please correct “E. Coli” in the y-axis titles. Line 427, please add references to support the claim. Figure 7, please explain “AUC”, “RFU” and ”RLU” in the figure caption. Line 517, “GO caused a broad, non-cell-specific triggering the production of all cytokines”. There seems to be a word missing here.

  1. Comment

The references in the text are in an unusual format in that the author names are not followed by "et al." in the cases where there is more than one author. This will not be a problem after formatting the references to the style of the journal Nanomaterials which is a number style. Also, the reference list is currently both numbered and in alphabetical order which is unconventional.

Response

References were formatted according to journal requirements

  1. Comment

In the end of the Introduction, the authors have listed the specific goals of the study. However, the general aim of the study has not been stated. It would be helpful to also articulate the overall scientific goal of the study in the end of the Introduction.

Response

Aim of the study was added in the end of the Introduction

  1. Comments

Line 87, it could be stated if the graphene oxide materials were purchased as powders or suspensions.

Line 92, “Sigma” should be “Sigma-Aldrich”.

It could be added that RPMI-1640 is cell culture medium.

Line 106, it could be specified if probe or bath sonication was used.

Line 107, the volume and concentration of Cl-CH2-COOH should be specified.

Line 108, please add how the washing was done.

In section 2.3. “Characterization of GO and GO-PEG”, in the cases where the characterization was done in suspensions, it should be specified which medium or water was used and what was the nanomaterial concentration.

In section 2.4. “Isolation of monocytes from human blood”, the source and circumstances of obtaining human blood for the study should be explained. Later in the manuscript 4 subjects are mentioned. This should be explained in the methods. Who were the individuals, were they healthy etc.

Line 141, “CO-PEG” should be “GO-PEG”

Response

Required experimental details were added in corresponding places of manuscript. Misprints and errors were fixed.

  1. Comment

Line 147, the method of SDS-PAGE should be explained. For example, how was the gel stained, were the protein band sizes quantified somehow (later the authors compare the amounts of proteins associated with different GO-PEG but quantitative analysis has not been described)

Response

Detailed description of SDS-PAGE was added. Protein quantification procedure was described in Methods, besides measurement procedure was illustrated with Figure S1. Quantities of protein in each SDS-PAGE lane expressed in arbitrary units were added in Figures 3 and S1.

  1. Comment

In section 2.6. “Uptake of GO-PEG by monocytes” it is mentioned that fluorescence was measured but it is unclear what was the source of fluorescence. Were the cells stained with a fluorescence stain? In the results it is mentioned that also side scatter was measured but this has not been described in the methods.

Response

Graphene oxide itself exhibits fluorescence [DOI: https://doi.org/10.1166/jbn.2011.1186]. In initial experiments, we found that on the CytoFLEX S instrument, the maximum difference between the MFI of monocytes and GO nanoparticles can be observed in the channel for Cy7 dye (488 nm laser, bandpass filter 780 ± 60 nm). Therefore, to assess the uptake of GO by monocytes, the percentage of cells fluorescing in this channel was determined, along with the median fluorescence and the engulfment index (median fluorescence of monocytes in samples with GO, divided by the number of monocytes). In addition, the percentage of highly granular (nanoparticle-engulfing) cells was determined using an SSC-A histogram. The linear gate was set according to the control without GO nanoparticles.

  1. Comment

Lines 164-165, specify how many cells were counted per sample and in how many replicates to obtain quantitative data by microscopy.

Response.

100 cells were counted per sample, and 2 replicates were taken. This information was added to the Section 2.7.

  1. Comment

Line 172, it is unclear, based on the composition of the staining buffer, what the stain was.

Response

In this case, there is no dye in the buffer. We just washed cells with phosphate buffer that is usually used to in the course cell staining. We have removed the word "staining" from the sentence for clarity.

  1. Comment

Line 201, LPS has not been explained.

Response

LPS stands for lipopolysaccharide. The meaning of abbreviation was added to the text.

  1. Comment

A separate subsection about the statistical analysis of the data, replicates etc. should be added to the Materials and Methods section.

Response

Section 2.11. “Statistical analysis” was added

  1. Comment

Lines 207-208, add references to the sentence to support this claim.

Response

Reference was added

  1. Comment

In the caption of Figure 2 and line 228, IR should be FTIR.

Response

Fixed

  1. The FTIR absorption bands and functional groups discussed in lines 227-237 should be indicated in the spectra in Figure 2. Otherwise, it is difficult to understand where these specific bands are because some regions contain multiple bands close together.

We changed graphics with FTIR results according to the Reviewer’s comments.

  1. Lines 264 and 271, “Table” should be “Table 1”.

Fixed

  1. Lines 267-268, the meaning of the sentence is unclear, please rephrase.

The sentence was changed

  1. Lines 268-269, has the TGA data for pristine GO presented? I did not notice this.

TGA curve for intact GO was added to Supplementary material (Figure S2)

  1. Lines 285-286, please add references to support the claim.

Reference was added

  1. Line 289, “human serum in concentrations of 5 and 25 µl”. This sentence needs to be corrected.

Changed to “human serum in concentrations of 5 and 25 µg/ml”

  1. Lines 297-298, “Branched PEG more efficiently shield nanoparticle surfaces…” This is not obvious from the comparison of SnGO and SbGO sample protein bands in Figure 3. Was any quantitative assessment done to compare the band sizes?

Semi-quantitative analysis of protein adsorbed on graphene oxide was performed by measuring the brightness of SDS-PAGE lanes using Image J software (details were added to the Section. 2.5.). Decrease of protein adsorption was observed for graphene oxide nanoparticles coated with branched PEG in comparison with those coated with linear PEG. Explanation was added to the section 3.2.

  1. Line 362, “HSA” has not been explained.

Abbreviation was changed to “human serum albumin”

  1. Line 365, “the amount of adsorbed did not…” There appears to be a word missing here.

Changed to “Nevertheless, the amount of adsorbed PROTEIN did not correlate with their cellular uptake.”

  1. Line 373, the meaning of “linear cells” is unclear.

Changed to “Cell lines”

  1. Figure 6, please correct “E. Coli” in the y-axis titles.

Fixed

  1. Line 427, please add references to support the claim.

Despite in is known that graphene oxide can both change ROS production by the immune cells [Yunus, M.A. et al.10.1007/s00005-021-00625-6] and affect their anti-pathogen activity such as phagocytosis or formation of neutrophil extracellular traps [Keshavan, S. et al. 10.1038/s41419-019-1806-8, Svadlakova, T. et al. 10.3390/nano11102510], there is still not enough experimental evidence that effects of graphene on ROS formation correlates with antimicrobial function of immune cells. Therefore, we changed the sentence to “Graphene family nanomaterials change ROS production in immune cells [Yunus, 2021]”.

  1. Figure 7, please explain “AUC”, “RFU” and ”RLU” in the figure caption.

Fixed

  1. Line 517, “GO caused a broad, non-cell-specific triggering the production of all cytokines”. There seems to be a word missing here.

The sentence was corrected